# Scalable Planning with Tensorflow for Hybrid Nonlinear Domains

**Ga Wu**                                    **Buser Say**

**Scott Sanner**
Department of Mechanical & Industrial Engineering, University of Toronto, Canada
email: {wuga,bsay,ssanner}@mie.utoronto.ca

## Abstract

Given recent deep learning results that demonstrate the ability to effectively optimize high-dimensional non-convex functions with gradient descent optimization on GPUs, we ask in this paper whether symbolic gradient optimization tools such as Tensorflow can be effective for planning in hybrid (mixed discrete and continuous) nonlinear domains with high dimensional state and action spaces? To this end, we demonstrate that hybrid planning with Tensorflow and RMSProp gradient descent *is* competitive with mixed integer linear program (MILP) based optimization on piecewise linear planning domains (where we can compute optimal solutions) and substantially outperforms state-of-the-art interior point methods for nonlinear planning domains. Furthermore, we remark that Tensorflow is highly scalable, converging to a strong plan on a large-scale concurrent domain with a total of 576,000 continuous action parameters distributed over a horizon of 96 time steps and 100 parallel instances in only 4 minutes. We provide a number of insights that clarify such strong performance including observations that despite long horizons, RMSProp avoids both the vanishing and exploding gradient problems. Together these results suggest a new frontier for highly scalable planning in nonlinear hybrid domains by leveraging GPUs and the power of recent advances in gradient descent with highly optimized toolkits like Tensorflow.

## 1   Introduction

Many real-world hybrid (mixed discrete continuous) planning problems such as Reservoir Control [Yeh, 1985], Heating, Ventilation and Air Conditioning (HVAC) [Erickson *et al.*, 2009; Agarwal *et al.*, 2010], and Navigation [Faulwasser and Findeisen, 2009] have highly nonlinear transition and (possibly nonlinear) reward functions to optimize. Unfortunately, existing state-of-the-art hybrid planners [Ivankovic *et al.*, 2014; Löhr *et al.*, 2012; Coles *et al.*, 2013; Piotrowski *et al.*, 2016] are not compatible with arbitrary nonlinear transition and reward models. While HD-MILP-PLAN [Say *et al.*, 2017] supports arbitrary nonlinear transition and reward models, it also assumes the availability of data to learn the state-transitions. Monte Carlo Tree Search (MCTS) methods [Coulom, 2006; Kocsis and Szepesvári, 2006; Keller and Helmert, 2013] including AlphaGo [Silver *et al.*, 2016] that can use any (nonlinear) black box model of transition dynamics do not inherently work with *continuous action spaces* due to the infinite branching factor. While MCTS with continuous action extensions such as HOOT [Weinstein and Littman, 2012] have been proposed, their continuous partitioning methods do not scale to high-dimensional continuous action spaces (for example, 100's or 1,000's of dimensions as used in this paper). Finally, offline model-free reinforcement learning (for example, Q-learning) with function approximation [Sutton and Barto, 1998; Szepesvári, 2010] and deep extensions [Mnih *et al.*, 2013] do not require any knowledge of the (nonlinear) transition model or reward, but they also do not directly apply to domains with high-dimensional continuous action spaces. That is, offline

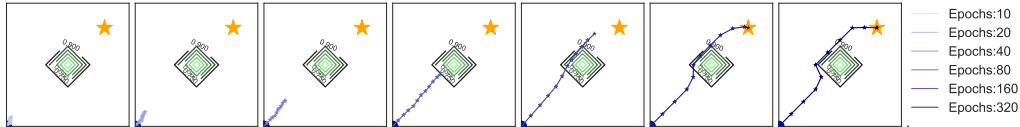

Figure 1: The evolution of RMSProp gradient descent based Tensorflow planning in a two-dimensional Navigation domain with nested central rectangles indicating nonlinearly increasing resistance to robot movement. (top) In initial RMSProp epochs, the plan evolves directly towards the goal shown as a star. (bottom) As later epochs of RMSProp descend the objective cost surface, the fastest path evolves to avoid the central obstacle entirely.

learning methods like Q-learning [Watkins and Dayan, 1992] require action maximization for every update, but in high-dimensional continuous action spaces such nonlinear function maximization is non-convex and computationally intractable at the scale of millions or billions of updates.

To address the above scalability and expressivity limitations of existing methods, we turn to Tensorflow [Abadi *et al.*, 2015], which is a symbolic computation platform used in the machine learning community for deep learning due to its compilation of complex layered symbolic functions into a representation amenable to fast GPU-based reverse-mode automatic differentiation [Linnainmaa, 1970] for gradient-based optimization. Given recent results in gradient descent optimization with deep learning that demonstrate the ability to effectively optimize high-dimensional non-convex functions, we ask whether Tensorflow can be effective for planning in discrete time, hybrid (mixed discrete and continuous) nonlinear domains with high dimensional state and action spaces?

Our results answer this question affirmatively, where we demonstrate that hybrid planning with Tensorflow and RMSProp gradient descent [Tieleman and Hinton, 2012] is surprisingly effective at planning in complex hybrid nonlinear domains[1]. As evidence, we reference figure 1, where we show Tensorflow with RMSProp efficiently finding and optimizing a least-cost path in a two-dimensional nonlinear Navigation domain. In general, Tensorflow with RMSProp planning results are competitive with optimal MILP-based optimization on piecewise linear planning domains. The performance directly extends to nonlinear domains where Tensorflow with RMSProp substantially outperforms interior point methods for nonlinear function optimization. Furthermore, we remark that Tensorflow converges to a strong plan on a large-scale concurrent domain with 576,000 continuous actions distributed over a horizon of 96 time steps and 100 parallel instances in 4 minutes.

To explain such excellent results, we note that gradient descent algorithms such as RMSProp are highly effective for non-convex function optimization that occurs in deep learning. Further, we provide an analysis of many transition functions in planning domains that suggest gradient descent on these domains will not suffer from either the vanishing or exploding gradient problems, and hence provide a strong signal for optimization over long horizons. Together these results suggest a new frontier for highly scalable planning in nonlinear hybrid domains by leveraging GPUs and the power of recent advances in gradient descent with Tensorflow and related toolkits.

## 2 Hybrid Nonlinear Planning via Tensorflow

In this section, we present a general framework of hybrid nonlinear planning along with a compilation of the objective in this framework to a symbolic recurrent neural network (RNN) architecture with action parameter inputs directly amenable to optimization with the Tensorflow toolkit.

### 2.1 Hybrid Planning

A hybrid planning problem is a tuple $\langle \mathcal{S}, \mathcal{A}, \mathcal{T}, \mathcal{R}, \mathcal{C} \rangle$ with $\mathcal{S}$ denoting the (infinite) set of hybrid states with a state represented as a mixed discrete and continuous vector, $\mathcal{A}$ the set of actions bounded by action constraints $\mathcal{C}$, $\mathcal{R} : \mathcal{S} \times \mathcal{A} \to \mathbb{R}$ the reward function and $\mathcal{T} : \mathcal{S} \times \mathcal{A} \to \mathcal{S}$ the transition

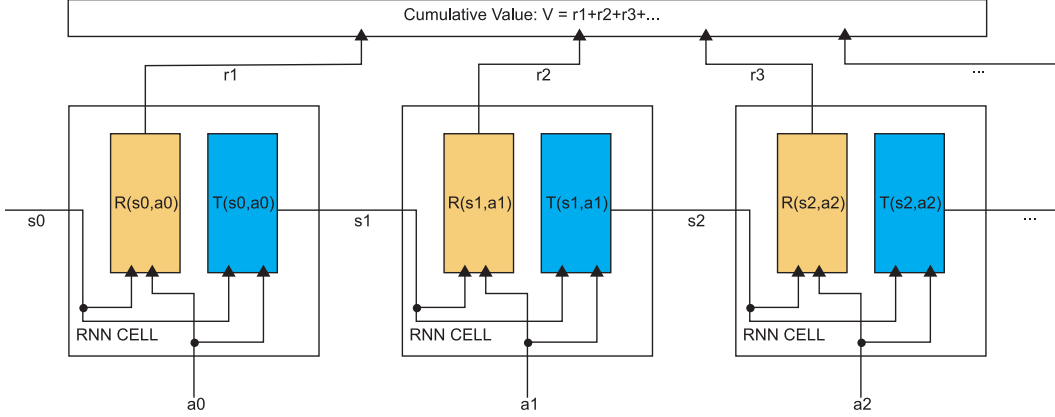

Figure 2: An recurrent neural network (RNN) encoding of a hybrid planning problem: A single-step reward and transition function of a discrete time decision-process are embedded in an RNN cell. RNN inputs correspond to the starting state and action; the outputs correspond to reward and next state. Rewards are additively accumulated in $V$. Since the entire specification of $V$ is a symbolic representation in Tensorflow with action parameters as inputs, the sequential action plan can be directly optimized via gradient descent using the auto-differentiated representation of V.

function. There is also an initial state $\mathbf{s}_0$ and the planning objective is to maximize the cumulative reward over a decision horizon of $H$ time steps. Before proceeding, we outline the necessary notation:

- $\mathbf{s}_t$: mixed discrete, continuous state vector at time $t$.
- $\mathbf{a}_t$: mixed discrete, continuous action vector at time $t$.
- $R(\mathbf{s}_t, \mathbf{a}_t)$: a non-positive reward function (i.e., negated costs).
- $T(\mathbf{s}_t, \mathbf{a}_t)$: a (nonlinear) transition function.
- $V = \sum_{t=1}^{H} r_t = \sum_{t=0}^{H-1} R(\mathbf{s_t}, \mathbf{a_t})$: cumulative reward value to maximize.

In general due to the stochastic nature of gradient descent, we will run a number of planning domain instances $i$ in parallel (to take the best performing plan over all instances), so we additionally define instance-specific states and actions:

- $s_{itj}$: the $j$th dimension of state vector of problem instance $i$ at time $t$.
- $a_{itj}$: the $j$th dimension of action vector of problem instance $i$ at time $t$.

## 2.2 Planning through Backpropagation

Backpropagation [Rumelhart *et al.*] is a standard method for optimizing parameters of large multilayer neural networks via gradient descent. Using the chain rule of derivatives, backpropagation propagates the derivative of the output error of a neural network back to each of its parameters in a single linear time pass in the size of the network using what is known as reverse-mode automatic differentiation [Linnainmaa, 1970]. Despite its relative efficiency, backpropagation in large-scale (deep) neural networks is still computationally expensive and it is only with the advent of recent GPU-based symbolic toolkits like Tensorflow [Abadi *et al.*, 2015] that recent advances in training very large deep neural networks have become possible.

In this paper, we reverse the idea of training parameters of the network given fixed inputs to instead optimizing the inputs (i.e., actions) subject to fixed parameters (effectively the transition and reward parameterization assumed *a priori* known in planning). That is, as shown in figure 2, given transition $T(\mathbf{s}_t, \mathbf{a}_t)$ and reward function $R(\mathbf{s}_t, \mathbf{a}_t)$, we want to optimize the input $\mathbf{a}_t$ for all $t$ to maximize the accumulated reward value $V$. Specifically, we want to optimize *all* actions $\mathbf{a} = (\mathbf{a}_1, \ldots, \mathbf{a}_{H-1})$ w.r.t. a planning loss $L$ (defined shortly) that we minimize via the following gradient update schema

$$\mathbf{a}' = \mathbf{a} - \eta \frac{\partial L}{\partial \mathbf{a}}, \tag{1}$$

where $\eta$ is the optimization rate and the partial derivatives comprising the gradient based optimization in problem instance $i$ are computed as

$$
\begin{aligned}
\frac{\partial L}{\partial a_{itj}} &= \frac{\partial L}{\partial L_i} \frac{\partial L_i}{\partial a_{itj}} \\
&= \frac{\partial L}{\partial L_i} \frac{\partial L_i}{\partial \mathbf{s}_{it+1}} \frac{\partial \mathbf{s}_{it+1}}{\partial a_{itj}} \\
&= \frac{\partial L}{\partial L_i} \frac{\partial \mathbf{s}_{it+1}}{\partial a_{itj}} \sum_{\tau=t+2}^{T} [\frac{\partial L_i}{\partial r_{i\tau}} \frac{\partial r_{i\tau}}{\partial \mathbf{s}_{i\tau}} \prod_{\kappa=\tau}^{t+2} \frac{\partial \mathbf{s}_{i\kappa}}{\mathbf{s}_{i\kappa-1}}].
\end{aligned}
\tag{2}
$$

We must now connect our planning objective to a standard Tensorflow loss function. First, however, let us assume that we have $N$ structurally identical instances $i$ of our planning domain given in Figure 2, each with objective value $V_i$; then let us define $\mathbf{V} = (\ldots, V_i, \ldots)$. In Tensorflow, we choose Mean Squared Error (MSE), which given two continuous vectors $\mathbf{Y}$ and $\mathbf{Y}^*$ is defined as $\mathrm{MSE}(\mathbf{Y}, \mathbf{Y}^*) = \frac{1}{N}\|\mathbf{Y}^* - \mathbf{Y}\|^2$. We specifically choose to minimize $L = \mathrm{MSE}(\mathbf{0}, \mathbf{V})$ with inputs of constant vector $\mathbf{0}$ and value vector $\mathbf{V}$ in order to maximize our value for each instance $i$; we remark that here we want to independently maximize each non-positive $V_i$, but minimize each positive $V_i^2$ which is achieved with MSE. We will further explain the use of MSE in a moment, but first we digress to explain why we need to solve multiple problem instances $i$.

Since both transition and reward functions are not assumed to be convex, optimization on a domain with such dynamics could result in a local minimum. To mitigate this problem, we use randomly initialized actions in a batch optimization: we optimize multiple mutually independent planning problem instances $i$ simultaneously since the GPU can exploit their parallel computation, and then select the best-performing action sequence among the independent simultaneously solved problem instances. MSE then has dual effects of optimizing each problem instance $i$ independently and providing fast convergence (faster than optimizing $V$ directly). We remark that simply defining the objective $V$ and the definition of all state variables in terms of predecessor state and action variables via the transition dynamics (back to the known initial state constants) is enough for Tensorflow to build the symbolic directed acyclic graph (DAG) representing the objective and take its gradient with respect to to all free action parameters as shown in (2) using reverse-mode automatic differentiation.

## 2.3 Planning over Long Horizons

The Tensorflow compilation of a nonlinear planning problem reflects the same structure as a recurrent neural network (RNN) that is commonly used in deep learning. The connection here is not superficial since a longstanding difficulty with training RNNs lies in the vanishing gradient problem, that is, multiplying long sequences of gradients in the chain rule usually renders them extremely small and make them irrelevant for weight updates, especially when using nonlinear transfer functions such as a sigmoid. However in hybrid planning problems, continuous state updates often take the form $s_{i(t+1)j} = s_{itj} + \Delta$ for some $\Delta$ function of the state and action at time $t$. Critically we note that the transfer function here is linear in $s_{itj}$ which is the largest determiner of $s_{i(t+1)j}$, hence avoiding vanishing gradients.

In addition, a gradient can explode with the chain rule through backpropagation if the elements of the Jacobian matrix of state transitions are too large. In this case, if the planning horizon is large enough, a simple Stochastic Gradient Descent (SGD) optimizer may suffer from overshooting the optimum and never converge (as our experiments appear to demonstrate for SGD). The RMSProp optimization algorithm has a significant advantage for backpropagation-based planning because of its ability to perform gradient normalization that avoids exploding gradients and additionally deals with piecewise gradients [Balduzzi *et al.*, 2016] that arise naturally as conditional transitions in many nonlinear domains (e.g., the Navigation domain of Figure 1 has different piecewise transition dynamics depending on the starting region). Specifically, instead of naively updating action $a_{itj}$ through equation 1, RMSProp maintains a decaying root mean squared gradient value $G$ for each variable, which averages over squared gradients of previous epochs

$$
G'_{a_{itj}} = 0.9 G_{a_{itj}} + 0.1 (\frac{\partial L}{\partial a_{itj}})^2,
\tag{3}
$$

and updates each action variable through

$$a'_{itj} = a_{itj} - \frac{\eta}{\sqrt{G_{a_{itj}} + \epsilon}} \frac{\partial L}{\partial a_{itj}}.\qquad(4)$$

Here, the gradient is relatively small and consistent over iterations. Although the Adagrad [Duchi *et al.*, 2011] and Adadelta [Zeiler, 2012] optimization algorithms have similar mechanisms, their learning rate could quickly reduce to an extremely small value when encountering large gradients. In support of these observations, we note the superior performance of RMSProp in Section 3.

## 2.4 Handling Constrained and Discrete Actions

In most hybrid planning problems, there exist natural range constraints for actions. To handle those constraints, we use projected stochastic gradient descent. Projected stochastic gradient descent (PSGD) is a well-known descent method that can handle constrained optimization problems by projecting the parameters (actions) into a feasible range after each gradient update. To this end, we clip all actions to their feasible range after each epoch of gradient descent.

For planning problems with discrete actions, we use a one-hot encoding for optimization purposes and then use a $\{0, 1\}$ projection for the maximal action to feed into the forward propagation. In this paper, we focus on constrained continuous actions which are representative of many hybrid nonlinear planning problems in the literature.

# 3 Experiments

In this section, we introduce our three benchmark domains and then validate Tensorflow planning performance in the following steps. (1) We evaluate the optimality of the Tensorflow backpropagation planning on linear and bilinear domains through comparison with the optimal solution given by Mixture Integer Linear Programming (MILP). (2) We evaluate the performance of Tensorflow backpropagation planning on nonlinear domains (that MILPs cannot handle) through comparison with the Matlab-based interior point nonlinear solver FMINCON. (4) We investigate the impact of several popular gradient descent optimizers on planning performance. (5) We evaluate optimization of the learning rate. (6) We investigate how other state-of-the-art hybrid planners perform.

## 3.1 Domain Descriptions

**Navigation:**  The Navigation domain is designed to test the ability of optimization of Tensorflow in a relatively small environment that supports different complexity transitions. Navigation has a two-dimensional state of the agent location $\mathbf{s}$ and a two-dimensional action $\mathbf{a}$. Both of state and action spaces are continuous and constrained by their maximum and minimum boundaries separately.

The objective of the domain is for an agent to move to the goal state as soon as possible (cf. figure 1). Therefore, we compute the reward based on the Manhattan distance from the agent to the goal state at each time step as $R(\mathbf{s}_t, \mathbf{a}_t) = -\|\mathbf{s}_t - \mathbf{g}\|_1$, where $\mathbf{g}$ is the goal state.

We designed three different transitions; from left to right, nonlinear, bilinear and linear:

$$d_t = \|\mathbf{s_t} - \mathbf{z}\|$$
$$\lambda = \frac{2}{1 + \exp(-2d_t)} - 0.99$$
$$\mathbf{p} = \mathbf{s}_t + \lambda \mathbf{a}_t$$
$$T(\mathbf{s}_t, \mathbf{a}_t) = \max(\mathbf{u}, \min(\mathbf{l}, \mathbf{p})),$$
$$(5)$$

$$d_t = \sum_{j=1}^{2} |s_{tj} - z_j|$$
$$\lambda = \begin{cases} \frac{d_t}{4}, & d_t < 4 \\ 1, & d_t \geq 4 \end{cases}$$
$$\mathbf{p} = \mathbf{s}_t + \lambda \mathbf{a}_t$$
$$T(\mathbf{s}_t, \mathbf{a}_t) = \max(\mathbf{u}, \min(\mathbf{l}, \mathbf{p})),$$
$$(6)$$

$$d_t = \|\mathbf{s}_t - \mathbf{z}\|_1$$
$$\lambda = \begin{cases} 0.8, & 3.6 \leq d_t < 4 \\ 0.6, & 2.4 \leq d_t < 3.6 \\ 0.4, & 1.6 \leq d_t < 2.4 \\ 0.2, & 0.8 \leq d_t < 1.6 \\ 0.05, & d_t < 0.8 \\ 1, & d_t \geq 4 \end{cases}$$
$$\mathbf{p} = \mathbf{s}_t + \lambda \mathbf{a}_t$$
$$T(\mathbf{s}_t, \mathbf{a}_t) = \max(\mathbf{u}, \min(\mathbf{l}, \mathbf{p})),$$
$$(7)$$

The nonlinear transition has a velocity reduction zone based on its Euclidean distance to the center $\mathbf{z}$. Here, $d_t$ is the distance from the deceleration zone $\mathbf{z}$, $\mathbf{p}$ is the proposed next state, $\lambda$ is the velocity reduction factor, and $\mathbf{u}, \mathbf{l}$ are upper and lower boundaries of the domain respectively.

The bilinear domain is designed to compare with MILP where domain discretization is possible. In this setting, we evaluate the efficacy of approximately discretizing bilinear planning problems into MILPs. Equation 6 shows the bilinear transition function.

The linear domain is the discretized version of the bilinear domain used for MILP optimization. We also test Tensorflow on this domain to see the optimality of the Tensorflow solution. Equation 7 shows the linear transition function.

**Reservoir Control:** Reservoir Control [Yeh, 1985] is a system to control multiple connected reservoirs. Each of the reservoirs in the system has a single state $s_j \in \mathbb{R}$ that denotes the water level of the reservoir $j$ and a corresponding action to permit a flow $a_j \in [0, s_j]$ from the reservoir to the next downstream reservoir.

The objective of the domain is to maintain the target water level of each reservoir in a safe range and as close to half of its capacity as possible. Therefore, we compute the reward through:

$$c_j = \begin{cases} 0, & L_j \leq s_j \leq U_j \\ -5, & s_j < L_j \\ -100, & s_j > U_j \end{cases}$$

$$R(\mathbf{s}_t, \mathbf{a}_t) = -\|\mathbf{c} - 0.1 * |\frac{(\mathbf{u} - \mathbf{l})}{2} - \mathbf{s_t}|\|_1,$$

where $c_j$ is the cost value of Reservoir $j$ that penalizes water levels outside a safe range.

In this domain, we introduce two settings: namely, Nonlinear and Linear. For the nonlinear domain, nonlinearity due to the water loss $e_j$ for each reservoir $j$ includes water usage and evaporation. The transition function is

$$\mathbf{e}_t = 0.5 * \mathbf{s}_t \odot sin(\frac{\mathbf{s}_t}{m}), \quad T(\mathbf{s}_t, \mathbf{a}_t) = \mathbf{s}_t + \mathbf{r}_t - \mathbf{e}_t - \mathbf{a}_t + \mathbf{a}_t \Sigma, \tag{8}$$

where $\odot$ represents an elementwise product, $\mathbf{r}$ is a rain quantity parameter, $m$ is the maximum capacity of the largest tank, and $\Sigma$ is a lower triangular adjacency matrix that indicates connections to upstream reservoirs.

For the linear domain, we only replace the nonlinear function of water loss by a linear function:

$$\mathbf{e}_t = 0.1 * \mathbf{s}_t, \quad T(\mathbf{s}_t, \mathbf{a}_t) = \mathbf{s}_t + \mathbf{r}_t - \mathbf{e}_t - \mathbf{a}_t + \mathbf{a}_t \Sigma, \tag{9}$$

Unlike Navigation, we do not limit the state dimension of the whole system into two dimensions. In the experiments, we use domain setting of a network with 20 reservoirs.

**HVAC:** Heating, Ventilation, and Air Conditioning [Erickson *et al.*, 2009; Agarwal *et al.*, 2010] is a centralized control problem, with concurrent controls of multiple rooms and multiple connected buildings. For each room $j$ there is a state variable $s_j$ denoting the temperature and an action $a_j$ for sending the specified volume of heated air to each room $j$ via vent actuation.

The objective of the domain is to maintain the temperature of each room in a comfortable range and consume as little energy as possible in doing so. Therefore, we compute the reward based through:

$$\mathbf{d}_t = |\frac{(\mathbf{u} - \mathbf{l})}{2} - \mathbf{s}_t|, \quad \mathbf{e}_t = \mathbf{a}_t * C, \quad R(\mathbf{s}_t, \mathbf{a}_t) = -\|\mathbf{e}_t + \mathbf{d}_t\|_1,$$

where $C$ is the unit electricity cost.

Since thermal models for HVAC are inherently nonlinear, we only present one version with a nonlinear transition function:

$$\boldsymbol{\theta}_t = \mathbf{a}_t \odot (F^{vent} - \mathbf{s}_t), \quad \boldsymbol{\phi}_t = (\mathbf{s}_t \mathbf{Q} - \mathbf{s}_t \odot \sum_{j=1}^{J} \mathbf{q}_j)/w_q$$

$$\boldsymbol{\vartheta}_t = (F_t^{out} - \mathbf{s}_t) \odot \mathbf{o}/w_o, \quad \boldsymbol{\phi}_t = (F_t^{hall} - \mathbf{s}_t) \odot \mathbf{h}/w_h$$

$$T(\mathbf{s}_t, \mathbf{a}_t) = \mathbf{s}_t + \alpha * (\boldsymbol{\theta}_t + \boldsymbol{\phi}_t + \boldsymbol{\vartheta}_t + \boldsymbol{\phi}_t), \tag{10}$$

where $F^{vent}$, $F_t^{out}$ and $F_t^{hall}$ are temperatures of the room vent, outside and hallway, respectively, $\mathbf{Q}$, $\mathbf{o}$ and $\mathbf{h}$ are respectively the adjacency matrix of rooms, adjacency vector of outside areas, and the adjacency vector of hallways. $w_q$, $w_o$ and $w_h$ are thermal resistances with a room and the hallway and outside walls, respectively.

In the experiments, we work with a building layout with five floors and 12 rooms on each floor for a total of 60 rooms. For scalability testing, we apply batched backpropagation on 100 instances of such domain simultaneously, of which, there are 576,000 actions needed to plan concurrently.

## 3.2 Planning Performance

In this section, we investigate the performance of Tensorflow optimization through comparison with the MILP on linear domains and with Matlab's fmincon nonlinear interior point solver on nonlinear domains. We ran our experiments on Ubuntu Linux system with one E5-1620 v4 CPU, 16GB RAM, and one GTX1080 GPU. The Tensorflow version is beta 0.12.1, the Matlab version is R2016b, and the MILP version is IBM ILOG CPLEX 12.6.3.

### 3.2.1 Performance in Linear Domains

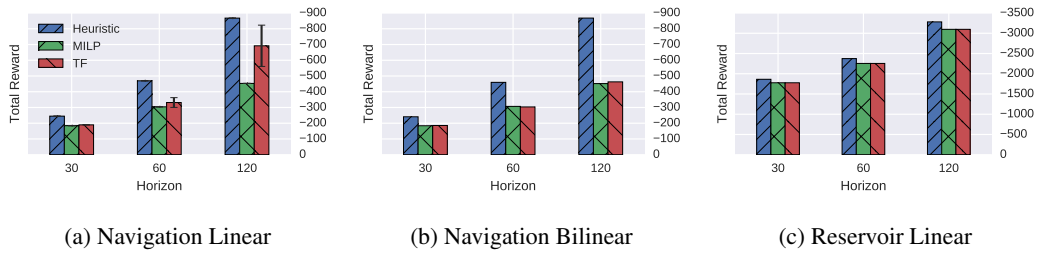

|     (a) Navigation Linear     |     (b) Navigation Bilinear     |     (c) Reservoir Linear     |

Figure 3: The total reward comparison (values are negative, lower bars are better) among Tensorflow (Red), MILP optimization guided planning (Green) and domain-specific heuristic policy (Blue). Error bars show standard deviation across the parallel Tensorflow instances; most are too small to be visible. The heuristic policy is a manually designed baseline solution. In the linear domains (a) and (c), the MILP is optimal and Tensorflow is near-optimal for five out of six domains.

In Figure 3, we show that Tensorflow backpropagation results in lower cost plans than domain-specific heuristic policies, and the overall cost is close to the MILP-optimal solution in five of six linear domains.

While Tensorflow backpropagation planning generally shows strong performance, when comparing the performance of Tensorflow on bilinear and linear domains of Navigation to the MILP solution (recall that the linear domain was discretized from the bilinear case), we notice that Tensorflow does much better relative to the MILP on the bilinear domain than the discretized linear domain. The reason for this is quite simple: gradient optimization of smooth bilinear functions is actually much easier for Tensorflow than the piecewise linear discretized version which has large piecewise steps that make it hard for RMSProp to get a consistent and smooth gradient signal. We additionally note that the standard deviation of the linear navigation domain is much larger than the others. This is because the *piecewise constant* transition function computing the speed reduction factor $\lambda$ provides a flat loss surface with no curvature to aid gradient descent methods, leading to high variation depending on the initial random starting point in the instance.

### 3.2.2 Performance in Nonlinear Domains

In figure 4, we show Tensorflow backpropagation planning always achieves the best performance compared to the heuristic solution and the Matlab nonlinear optimizer fmincon. For relatively simple domains like Navigation, we see the fmincon nonlinear solver provides a very competitive solution, while, for the complex domain HVAC with a large concurrent action space, the fmincon solver shows a complete failure at solving the problem in the given time period.

In figure 5(a), Tensorflow backpropagation planning shows 16 times faster optimization in the first 15s, which is close to the result given by fmincon at 4mins. In figure 5(b), the optimization speed of

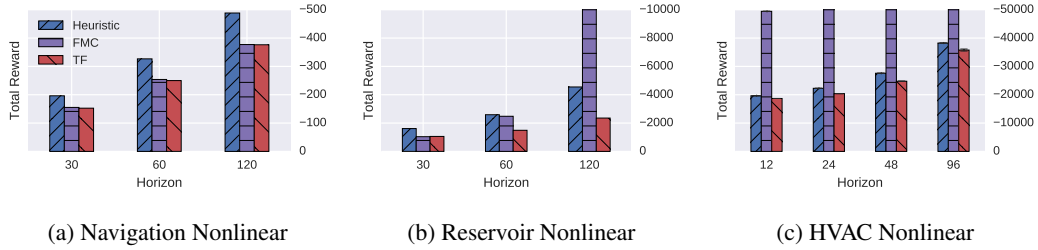

| (a) Navigation Nonlinear | (b) Reservoir Nonlinear | (c) HVAC Nonlinear |

Figure 4: The total reward comparison (values are negative, lower bars are better) among Tensorflow backpropagation planning (Red), Matlab nonlinear solver fmincon guided planning (Purple) and domain-specific heuristic policy (Blue). We gathered the results after 16 minutes of optimization time to allow all algorithms to converge to their best solution.

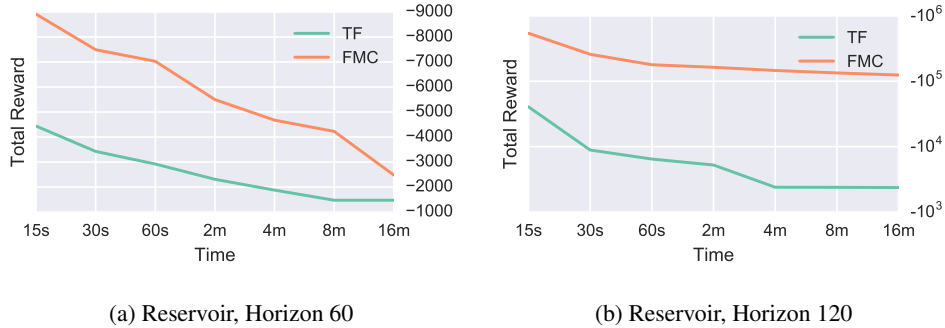

| (a) Reservoir, Horizon 60 | (b) Reservoir, Horizon 120 |

Figure 5: Optimization comparison between Tensorflow RMSProp gradient planning (Green) and Matlab nonlinear solver fmincon interior point optimization planning (Orange) on Nonlinear Reservoir Domains with Horizon (a) 60 and (b) 120. As a function of the logarithmic time x-axis, Tensorflow is substantially faster and more optimal than fmincon.

Tensorflow shows it to be hundreds of times faster than the fmincon nonlinear solver to achieve the same value (if fmincon does ever reach it). These remarkable results demonstrate the power of fast parallel GPU computation of the Tensorflow framework.

### 3.2.3  Scalability

In table 1, we show the scalability of Tensorflow backpropagation planning via the running times required to converge for different domains. The results demonstrate the extreme efficiency with which Tensorflow can converge on exceptionally large nonlinear hybrid planning domains.

| Domain | Dim | Horizon | Batch | Actions | Time |
|--------|-----|---------|-------|---------|------|
| Nav. | 2 | 120 | 100 | 24000 | $< 1$mins |
| Res. | 20 | 120 | 100 | 240000 | 4mins |
| HVAC | 60 | 96 | 100 | 576000 | 4mins |

Table 1: Timing evaluation of the largest instances of the three domains we tested. All of these tests were performed on the nonlinear versions of the respectively named domains.

### 3.2.4  Optimization Methods

In this experiment, we investigate the effects of different backpropagation optimizers. In figure 6(a), we show that the RMSProp optimizer provides exceptionally fast convergence among the five standard optimizers of Tensorflow. This observation reflects the previous analysis and discussion concerning equation (4) that RMSProp manages to avoid exploding gradients. As mentioned, although Adagrad and Adadelta have similar mechanisms, their normalization methods may cause vanishing gradients after several epochs, which corresponds to our observation of nearly flat curves for these methods. This is a strong indicator that exploding gradients are a significant concern for hybrid planning with gradient descent and that RMSProp performs well despite this well-known potential problem for gradients over long horizons.

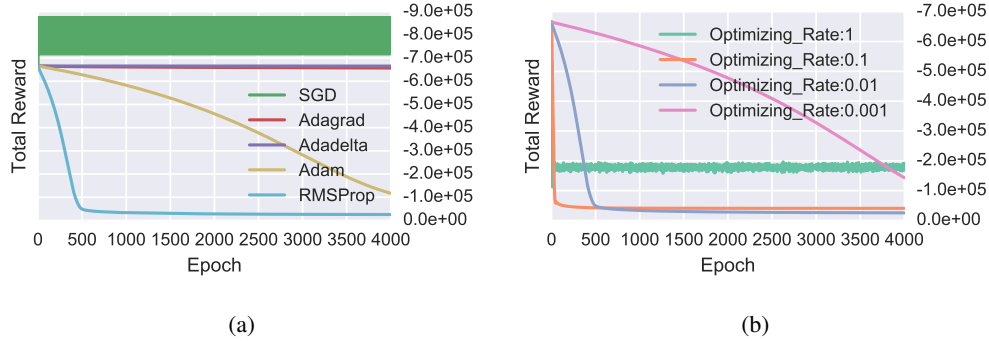

(a)                                          (b)

Figure 6: (a) Comparison of Tensorflow gradient methods in the HVAC domain. All of these optimizers use the same learning rate of 0.001. (b) Optimization learning rate comparison of Tensorflow with the RMSProp optimizer on HVAC domain. The optimization rate 0.1 (Orange) gave the fastest initial convergence speed but was not able to reach the best score that optimization rate 0.001 (Blue) found.

### 3.2.5 Optimization Rate

In figure 6(b), we show the best learning optimization rate for the HVAC domain is 0.01 since this rate converges to near-optimal extremely fast. The overall trend is smaller optimization rates have a better opportunity to reach a better final optimization solution, but can be extremely slow as shown for optimization rate 0.001. Hence, while larger optimization rates may cause overshooting the optima, rates that are too small may simply converge too slowly for practical use. This suggests a critical need to tune the optimization rate per planning domain.

### 3.3 Comparison to State-of-the-art Hybrid Planners

Finally, we discuss and test the scalability of the state-of-art hybrid planners on our hybrid domains. We note that neither DiNo [Piotrowski *et al.*, 2016], dReal [Bryce *et al.*, 2015] nor SMTPlan [Cashmore *et al.*, 2016] support general metric optimization. We ran ENHSP [Scala *et al.*, 2016] on a much smaller version of the HVAC domain with only 2 rooms over multiple horizon settings. We found that ENHSP returned a feasible solution to the instance with horizon equal to 2 in 31 seconds, whereas the rest of the instances with greater horizon settings timed out with an hour limit.

## 4 Conclusion

We investigated the practical feasibility of using the Tensorflow toolbox to do fast, large-scale planning in hybrid nonlinear domains. We worked with a direct symbolic (nonlinear) planning domain compilation to Tensorflow for which we optimized planning actions directly through gradient-based backpropagation. We then investigated planning over long horizons and suggested that RMSProp avoids both the vanishing and exploding gradient problems and showed experiments to corroborate this finding. Our key empirical results demonstrated that Tensorflow with RMSProp is competitive with MILPs on linear domains (where the optimal solution is known — indicating near optimality of Tensorflow and RMSProp for these non-convex functions) and strongly outperforms Matlab's state-of-the-art interior point optimizer on nonlinear domains, optimizing up to 576,000 actions in under 4 minutes. These results suggest a new frontier for highly scalable planning in nonlinear hybrid domains by leveraging GPUs and the power of recent advances in gradient descent such as RMSProp with highly optimized toolkits like Tensorflow.

For future work, we plan to further investigate Tensorflow-based planning improvements for domains with discrete action and state variables as well as difficult domains with only terminal rewards that provide little gradient signal guidance to the optimizer.

## Footnotes

[1]The approach in this paper is implemented in Tensorflow, but it is not specific to Tensorflow. While "scalable hybrid planning with symbolic representations, auto-differentiation, and modern gradient descent methods for non-convex functions implemented on a GPU" would make for a more general description of our contributions, we felt that "Tensorflow" succinctly imparts at least the spirit of all of these points in a single term.

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
