[Reviews · NeurIPS 2017]

Reviewer 1



This paper documents an approach to planning in domains with hybrid state and action spaces, using efficient stochastic gradient descent methods, in particular, in this case, as implemented in TensorFlow. Certainly, the idea of optimizing plans or trajectories using gradient methods is not new (lots of literature on shooting methods, using fmincon, etc. exists). And, people have long understood that random restarts for gradient methods in non-linear problems are a way to mitigate problems with local optimal. What this paper brings is (1) the idea of running those random restarts in parallel and (b) using an existing very efficient implementation of SGD. I'm somewhat torn, because it seems like a good idea, but also not very surprising. So, I'm not exactly sure how novel it is. My major concern about the paper, though, is how it deals with discrete dimensions of the state and action and, more generally, a lack of differentiability in the transition or reward model. All of the example domains seem to have dynamics that are continuous (though not differentiable). It seems reasonably clear that a method such as this would not work well on a completely discrete domain. It would be important, if this paper gets published to spend more time exploring exactly how much non-continuity affects the ability of the method to solve problems. The next biggest concern has to do with reward: although the paper discusses ways of mitigating the vanishing gradient problem, all of the domains presented here have a reward function that is non-zero at every stage and that almost surely provides a powerful gradient for the optimization algorithm. It would be important to discuss the effectiveness of this methods on domains in which there is only a terminal reward, for example. I found the comparison to RNNs to be somewhat distracting. Either I misunderstood, or the optimization algorithm is applied over a finite fixed horizon I"m sort of surprised by the statement that transition dynamics are typically non-stationary; I would have thought that in most planning problems they are, in fact, stationary. But I guess it doesn't matter too much. I was also somewhat confused by the assertion that the dynamics would typically have an additive-update form, where the additive component depends on the state and action: that doesn't actually constitute any form of restriction, does it? The comment about linearity in the update and vanishing gradients also does not make sense to me if delta is allowed to depend on s. Ultimately, I would say that the planning instances you do address are impressively large and the performance of the algorithm seems good. But it would be better to state more clearly what types of planning domains your method could really be expected to perform well on. ---------------------- After rebuttal: I appreciate the clarifications and think the paper would be stronger if you were somewhat clearer up front about exactly what subset of hybrid planning problems your method is appropriate for. For instance, the idea that such systems are "usually" non-stationary, or additive isn't really right; you can say that previous methods have focused on those, and that they constitute an important subclass, etc., etc. (which I believe), but "usually" just sounds wrong to me. Similarly, a clear discussion about what assumptions you're making about the reward function and what would happen if they were violated would also be important.

Reviewer 2



In this paper the authors proposed a method that is based on symbolic gradient optimization tools such as Tensorflow to perform effective planning with hybrid nonlinear domains and high dimensional states and actions. They argued that the hybrid planning tool based on Tensorflow and RMSprop is competitive with MILP based optimization, and it is highly scalable. In general, I found the idea of solving hybrid planning problem with symbolic gradient optimization tools interesting. However, while this paper focuses on empirical studies using the RNN reformulation of the long term planning problem, I found the contribution of this paper insufficient to meet the standard of a NIPS paper. Pros: 1) The idea of formulating a planning problem by reversing the training parameters of the network given fixed inputs to optimizing the inputs (i.e., actions) subject to fixed parameters (effectively the transition and reward parameterization assumed a priori known in planning) is very neat. In this case, the proposed method embedded the reward and state transition of each stage of the planning problem into a RNN cell, and the cumulative reward of the fixed horizon problem is reconstructed using the MSE objective function of the Tensorflow loss function. 2) Through imposing the specific form of the transition function in page 4, the authors argued that the vanishing gradient issue in the cascade RNN structure is avoided. 3) To handle constrained actions, the author proposed using projected stochastic gradient descent method to optimize for the actions, and to handle discrete actions, the author proposed using a one-hot encoding method to the inputs. These are reasonable and common strategies to handle constraints and discrete inputs. 4) Since computation in symbolic optimization packages such as Tensorflow is highly scalable with the recent advance in GPU computations, the proposed method is effective compared to MILP based methods. Cons: 1) Since the method aimed to optimize the inputs (which are the point-based actions), it is not solving a closed-loop optimal control problem where the solution is a “policy” (which is a functional mapping from state-action history to actions). I can see that this Tensorflow reformulation solves the open-loop control problem, which is a very restricted form of planning. 2) While the experiment section contains three domains, the transition probability and the reward functions are rather specific. Especially the transition of the second and the third experiments follow the linear assumption that avoids the vanishing gradient issue in the cascade RNN structure. This leave me unconvinced that the RNN reformulation trick work for a general open-loop planning problem. 3) While the RNN reformulation of the planning problem is parallelizable, explicit performance analysis (in terms of speed and sub-optimality trade-offs) and its comparisons with MILP based method are not provided. Conclusion: While I very much appreciate the extended numerical experiments performed by the authors. These experiments are too simplified and specific. In my opinion it is insufficient to convince readers that the proposed method outperforms MILP based solution algorithms in a general setting. Furthermore, numerical complexities and performance bounds in terms of sub-optimality (or asymptotic optimality) are omitted. Unless a more detailed analysis on the algorithms is given, I unfortunately do not think this paper has sufficient materials to pass the acceptance threshold.

Reviewer 3



This paper describes a gradient ascent approach to action selection for planning domains with both continuous and discrete actions. The paper shows that backpropagation can be used to optimize the actions while avoiding long horizon problems such as exploding or vanishing gradients. Experimental results are given for 2D planar navigation, reservoir control and HVAC control and the proposed approach is shown to outperform existing hybrid optimization techniques. Planning in hybrid domains is an important problem and existing methods do not scale to real world domains. This paper proposes a scalable solution to these problems and demonstrates its usefulness on several domains including two real world domains. As such, I believe it makes an important contribution to addressing real world planning problems. The paper is well written except for some minor comments (see below). Evidence that the proposed solution works comes in the form of several experiments that demonstrate scalability and solution quality. I have two concerns about the experiments that I would like to see the author's comments on: 1. The paper lacks any statistical significance results, error bars, or other statistics of variation. 2. Are the experimental baselines the state-of-the-art of hybrid domain planning problems? The introduction mentions other methods but justification is not given for not making an empirical comparison. This paper needs a distinct related work section. The current treatment of related work is limited and is just in the introduction. My last major question is whether or not Tensorflow is a critical part of the proposed method? It seems that what is important is the use of backpropagation and Tensorflow is just an implementation detail. I think the impact of the work might be more if the method was presented more generally. Or is there something I'm missing that makes Tensorflow essential? Overall, the work presents a useful method for planning in hybrid domain problems. Its main limitations are the handling of related literature and statistical significance. Clarifying these issues and addressing the minor comments given below would help improve the impact of the work. I have read the author's response and other reviews. Minor comments: Line 35: I.e., -> i.e., Line 35: Should cite Watkins for Q-learning Line 106: Do you mean optimizing V directly? Line 117: What do you mean by transfer function? Did you mean activation function? Line 134: This statement seems too strong? Why are there always constraints on the actions? Is it more appropriate to say, "planning problems frequently have constraints on the actions that can be taken"? Line 164 - 167: The top line (d_t = ||s_t - s||) is too far over. Line 170: It is not clear what the dynamics are for linear and bilinear. It would be useful to write them explicitly as done for the nonlinear dynamics. Figure 3: This figure would be more clear if the y-axis was in the correct direction. It would also be useful to have a statement such as "lower is better" so the reader can quickly compare TF to the other baselines. Figure 3: Since the MILP solution is optimal it would be more useful to see relative error values for TF and Heuristic. Absolute reward isn't as informative since its sensitive to reward scaling. All figures: It would be nice to see error bars on all applicable plots Line 235: "to for" -> for Figure 6: space before (b) All figures: check for black and white printing, especially figure 6 Line 262: RMSProb -> RMSProp